# Bots’ Activity on COVID-19 Pro and Anti-Vaccination Networks: Analysis of Spanish-Written Messages on Twitter

**DOI:** 10.3390/vaccines10081240

**Published:** 2022-08-02

**Authors:** Carlos Ruiz-Núñez, Sergio Segado-Fernández, Beatriz Jiménez-Gómez, Pedro Jesús Jiménez Hidalgo, Carlos Santiago Romero Magdalena, María del Carmen Águila Pollo, Azucena Santillán-Garcia, Ivan Herrera-Peco

**Affiliations:** 1PhD Program in Biomedicine, Translational Research and New Health Technologies, School of Medicine, University of Malaga, Blvr. Louis Pasteur, 29010 Málaga, Spain; carlos.ruiz@uma.es; 2Nursing Department, Faculty of Medicine, Universidad Alfonso X el Sabio, Avda Universidad, 1, Villanueva de la Cañada, 28691 Madrid, Spain; ssegafer@uax.es (S.S.-F.); bgomejim@uax.es (B.J.-G.); maguipol@uax.es (M.d.C.Á.P.); 3Traumatology and Orthopedic Surgery Service, Hospital Universitario de Móstoles, C/Dr. Luis Montes s/n., 28935 Madrid, Spain; pjimenezh@salud.madrid.org; 4Faculty of Health Sciences, Universidad Alfonso X el Sabio, Avda Universidad, 1, Villanueva de la Cañada, 28691 Madrid, Spain; cromemag@uax.es; 5Valencia International University, C/Pintor Sorolla 21, 46002 Valencia, Spain; azucena.santillan@campusviu.es

**Keywords:** bots, COVID-19, misinformation, public health, social media, vaccines

## Abstract

This study aims to analyze the role of bots in the dissemination of health information, both in favor of and opposing vaccination against COVID-19. Study design: An observational, retrospective, time-limited study was proposed, in which activity on the social network Twitter was analyzed. Methods: Data related to pro-vaccination and anti-vaccination networks were compiled from 24 December 2020 to 30 April 2021 and analyzed using the software NodeXL and Botometer. The analyzed tweets were written in Spanish, including keywords that allow identifying the message and focusing on bots’ activity and their influence on both networks. Results: In the pro-vaccination network, 404 bots were found (14.31% of the total number of users), located mainly in Chile (37.87%) and Spain (14.36%). The anti-vaccination network bots represented 16.19% of the total users and were mainly located in Spain (8.09%) and Argentina (6.25%). The pro-vaccination bots generated greater impact than bots in the anti-vaccination network (*p* < 0.000). With respect to the bots’ influence, the pro-vaccination network did have a significant influence compared to the activity of human users (*p* < 0.000). Conclusions: This study provides information on bots’ activity in pro- and anti-vaccination networks in Spanish, within the context of the COVID-19 pandemic on Twitter. It is found that bots in the pro-vaccination network influence the dissemination of the pro-vaccination message, as opposed to those in the anti-vaccination network. We consider that this information could provide guidance on how to enhance the dissemination of public health campaigns, but also to combat the spread of health misinformation on social media.

## 1. Introduction

By the end of July 2022, according to data from the World Health Organization, more than 575 million people will have been infected worldwide by SARS-CoV-2, with 6.7 million people dying. Focusing on vaccines, 62.3% of the population is fully vaccinated and 67.9% of the world’s population is vaccinated with at least one dose.

During health emergencies, such as the one resulting from the COVID-19 pandemic, the need for information becomes a concern for many people [1,2]. Demand for information together with the large amount of information generated, which has led to an infodemic, considered a major health problem [3] globally, are associated with the emergence of uncertainty about what is or is not verified health information [4], causing a decrease in adherence to the recommendations of health authorities [5] but also making people less critical of the information consulted and, therefore, more prone to believe in biased information [6] or even misleading information [2].

This health disinformation and its rapid dissemination, coming from social media, significantly affects proper public health communication and decreases preventive measures [7]. The easy access to social media and the lack of control over the content generated mean that they can be considered a quick channel for the spread of unverified health information [8], representing a potential threat to public health [9], due to unverified health information exposure causing significant concerns, such as the generation of misinformation about treatments or modifying health care habits [3,6,10].

Health disinformation is developed both by human users and by automated accounts, controlled by a mathematical algorithm, and commonly known as “bots” [11]. These impersonate human users on social networks, such as Facebook, Twitter, Instagram, etc., focusing on the generation of contents, dissemination of disinformation, etc. [11,12].

Bots have been identified as key elements in the propagation of health disinformation [13], with great importance in relation to disinformation about vaccination [14,15]. In this regard, it is important to point out that, for instance, on the social network Twitter, fake news spreads faster than real news, since the former is mostly topical and triggers emotional reactions [16].

The bots used in these actions are mainly classified into two categories: (i) Social bots, those that automatically produce content and interact with human users in social networks, their goal being to modify the behavior and emotions of human users with regards to a specific topic [17,18]. (ii) Cyborgs or hybrid bot–human accounts, which also generate automated content like the social bot and seek to modify the behavior and emotions of human users on a specific subject, although presenting a flexible structure and adapting to conversations with other users [7,19].

Social bots can go unnoticed for social media users, for they are designed to have a similar appearance to human profiles (e.g., displaying a personal picture and stating a name or location) and behave online in a human-like manner (retweeting, quoting, or endorsing others’ posts or tweets) [17]. Setting up social bots does not require complex software or programming skills. There exist online forums, which provide easy and free instructions for implementing social bots, thus, facilitating their creation and management [14].

Bots are best known for the dissemination of information, based on political [17] or economic content [18]. Their involvement in public health information is not extensively widespread, although they are considered dangerous agents for the dissemination of verified health information and they promote misinformation amongst the population [14]. In addition, groups, such as the so-called anti-vaccine groups, habitually use these social bots in such a way that they can boost and spread their information further and faster than verified health information is transmitted by organizations and users [19,20].

In this context, the main objective of the present study was to analyze the role of bots in the dissemination of health information related to COVID-19 vaccines, both in favor of and against the vaccination policy.

## 2. Materials and Methods

### 2.1. Study Design and Ethics

An observational, retrospective, time-limited study was proposed, in which activity on the social network Twitter was analyzed.

Since this study is performed on a social network and only activity among Twitter users is measured, no approval from a Research Ethics Committee is required. However, accounts of individual users were anonymized in order to develop good research practices on social media [21].

### 2.2. Data Collection

The information from the tweets was extracted through an API (Application Programming Interface) search tool, using the professional version of the software NodeXL (Social Media Research Foundation). This application connects to the chosen social network and allows us to study and download information on dates, users, keywords, and even studies how the different entities are related, i.e., the influence and communication in that network of users shown in communication nodes.

To achieve the objectives proposed in this study, the Twitter users included in the data analysis were those who had sent tweets with the following features: (i) tweets published in Spanish; (ii) tweets containing keywords or hashtag (selected for their importance with Google Trends) pro-vaccination (“yomevacuno” #yomevacuno ”COVID-19” #COVID-19) or anti-vaccination hashtags (#yonomevacuno “yonomevacuno” ”COVID-19” #COVID-19) against COVID-19; and (iii) tweets that were published between 24 December 2020 (00:00 a.m. CET) and 30 April 2021 (23:59 p.m. CET).

To obtain data on the bot score we used the Botometer API V4 (Observatory on Social Media and the Network Science Institute at Indiana University, Indiana, USA) to compute bot scores of users. The value obtained ranged from 0.0 to 1.0; scores closer to 1 represent a higher chance of bot-ness, while those with scores closer to 0 probably belong to humans. Congruent with the sensitivity settings of previous studies [22,23], we set the threshold to 0.76 considering that values larger than or equal to 0.76 meant that the Twitter account is a bot; otherwise, the user was considered human [22]. In some cases, Botometer fails to output a score due to issues including account suspensions and authorization problems. Those accounts that could not be assessed were excluded from further analysis.

### 2.3. Data Analysis

The analysis of the data compiled was performed in several steps (Figure 1). The first step was to analyze the most influential Twitter users who employed the analyzed hashtags, as well as their characteristics, using the Betweenness Centrality Score (BCS), which measures the influence of a vertex over the flow of information to other vertices, always assuming that information will travel through the shortest vertex path. The BSC value reflects how a user can control the information, choosing whether to share it or not, disclosing it to their network [20,24]. Secondly, we analyzed the activity in pro and anti-vaccination networks. It enabled us to identify content, activities, and/or influential users that can be strongly associated with overall Twitter activity, measured by the metrics of interactions and impressions. The interactions were defined as ‘favorite’ and ‘retweets’; meanwhile, the impression is an indicator of propagation of information, obtained when the number of tweets is multiplied by the number of followers [25]. Third, a content analysis was performed with the categories created after analyzing the data. It is important to note that, in this category analysis, only original tweets were taken into account, since these were considered to be those that generated the actual content disseminated throughout the user network. The content and category coding were performed independently by two researchers and corroborated by a third person, whereby any differences in approach and focus were always discussed and resolved with full agreement.

Finally, for data analysis, descriptive and inferential statistics were used via the Statistical Package for the Social Sciences software (SPSS) version 23.0 (IBM, Armonk, NY, USA). First, the normality distribution of data was tested with Kolmogorov–Smirnov’s test and homoscedasticity with Levene’s test. Differences between groups were assessed with Student’s *t*-test and a two-sided *p*-value of 0.05 was considered statistically significant.

## 3. Results

### 3.1. User Analysis of Pro and Anti-Vaccination Networks

Within the pro-vaccination network, a total of 2823 unique users was observed and 240 users were eliminated due to not granting access to check the account. Finally, we obtained 2583 users, of which 162 were categorized as bots (6.27%) (Table 1), whereas, within the anti-vaccination network, 5039 users were found, of which 376 were eliminated due to not granting access to check the account. Finally, we obtained 4663 users, of which 420 were categorized as bots (9%) (Table 1).

We observed that, in relation to the geolocation, there are differences between both networks, where 65.59% out of the total (n = 265) in the pro-vaccination network had a defined geolocation in their profile, compared to 20.77% (n = 113) with real geolocation in the anti-vaccination network. Main countries where bots were observed were Chile (n = 153; 37.87%), followed by Spain (n = 58; 14.36%) in the pro-vaccination network. Regarding the anti-vaccination network, 8.09% (n = 44) out of the total were located in Spain, followed by 6.25% (n = 34) located in Argentina (Figure 2).

In relation to the interactions generated in each network, it was observed that 979,418 interactions were generated in the pro-vaccination network, of which 42,135 (4.3%) corresponded to bots, with a ratio between interactions and messages in this network of 136.6. The average number of interactions generated by these bots was 125.39.

In the anti-vaccination network, 97,875 interactions were generated, of which bots accounted for 4.41% (4724). The ratio between interactions and messages in this network is 0.65. The average number of interactions generated by these bots was 24.016 (Table 1).

Finally, the impressions generated in each network were analyzed. It was found that in the pro-vaccination network, 101,668,395 impressions took place, of which the bots generated 4,856,946 (4.75%). In the anti-vaccination network, a total of 6,032,115,787 impressions was generated, with 460,521,238 (7.63%) attributed to bots. (Table 1). When analyzing the effect of bots on impressions, it is found that their participation is significant in none of the networks (Table 1).

In the pro-vaccination network, users identified as humans were found to generate a higher number of interactions than bots on the network (*t* = 3.44; *p* < 0.05). However, no significant differences were observed when an analysis of the impressions was performed. In the anti-vaccination network, we found the same behavior as in the pro-vaccine network, with significant differences in interactions and no significant differences in interactions.

### 3.2. Behavior of Pro-Vaccination and Anti-Vaccination Networks

When both networks are compared, we can observe that the interactions differ significantly (*t* = −33.512; *p* < 0.05), with a greater effect in the pro-vaccination network (mean = 368.24) than in the anti-vaccination network (mean = 23.01). However, the anti-vaccine network showed a significant difference when impressions (*t* = 4,73; *p* < 0.05) and messages (*t* = 30.68; *p* < 0.05) were compared with the pro-vaccination network.

In relation to the behavior of bots in pro-vaccination and anti-vaccination networks, the possible effect of bots in the two networks was studied, finding that the number of messages and the interactions produced by the bots in both networks differed significantly (*t* = −33.512; *p* < 0.05), observing that bots had a greater effect in the pro-vaccination network than in the anti-vaccination network. When analyzing the number of messages sent by the bots in both networks, it was observed that the bots sent a greater number of messages (mean = 17.33) in the anti-vaccination network compared to the messages (mean = 1.91) in the pro-vaccination network (*t* = 9.01; *p* < 0.05). Subsequently, the role of bots versus human users was compared in each network. With respect to the pro-vaccination network, it was found that bots generated a significant impact on interactions (*t* = 30.571; *p* < 0.05), while this effect was not observed on impressions (*t* = 0.323; *p* = 0.46). Within the anti-vaccination network, no significant impact on impressions was found, being human users the most influential users (*t* = 4.198; *p* ≤ 0.05), while this effect was not observed on interactions (*t* = 0.22; *p* = 0.826) for bots.

### 3.3. Influence of Bots and Content Analysis

The most influential bots in both networks were analyzed and categorized using the BCS score (Table 2). It was found that, within the 20 most influential users in the pro-vaccination network, 3 were categorized as bots, placing 2 of them among the 5 most influencing users of that network: users PV1 and PV2. PV1 introduces itself as a citizen focused on political activism, PV2 as a citizen without further explanation, and PV3 as an alternative communication channel.

Within the anti-vaccination network, among the 20 users with the highest BSC value, there is only one user recognized as a bot, labeled as AV1, who presents itself as a former teacher.

Within the most influential bots in both networks, for the anti-vaccination network, a total of 15 messages was generated, all being tweets, sent by 14 users and accounting for 1525 interactions and 57,492 impressions. Within the pro-vaccination network, 11 messages posting topics against COVID-19 vaccination were detected, accounting for 71 interactions and 99,945 impressions.

The entire sample of bots’ messages was collected within the anti-vaccine and pro-vaccines networks and analyzed. Considering the different approaches within the pro and anti-vaccination networks, movement was analyzed in order to categorize them (Table 3), where we can see a predominance of certain tendencies. In the pro-vaccination stream, the political focus was the main group (51.85%), followed by general tweets not expressing a view or clear opinion (29.63%), also showing support for the anti-vaccine movement (7.41%).

Analysis of the anti-vaccination network shows a high prevalence of messages that include exclusive hashtags or images (33.02%), followed by anti-vaccine messages (20.93%). Finally, the third group of messages was defined by messages with political content (Table 3). The anti-vaccine messages were defined by three main themes: vaccine safety (70.69%), vaccine efficacy (6.89%), and beliefs about COVID-19 vaccine (22.42%).

## 4. Discussion

In this study, we analyzed both the presence of bots and their impact on the networks in which they operate, disseminating information about health topics, such as vaccination against COVID-19.

Firstly, we did not find a large number of user accounts that were identified as bots, a similar situation previously described in the literature, which states that the presence of bots on Twitter can be different for networks focused on social or political information, with the percentage of bots ranging from 7.1% to 9.9% [11] and even increasing in some cases, depending on the author, from 9% to 15% [26]; in networks focused on health information, values between 3.9% [13], 5.4% [27], and 6% [28] are observed. As can be seen in the networks analyzed in this study, the percentage of bots detected in the two networks studied can be associated to disinformation on health issues, such as vaccination. However, it is important to point out that, in each network, there exist a small number of bots disseminating messages contrary to the general thread of the network in which they are located.

Secondly, it is observed that the behavior of the bots does not differ, regardless of the network in which they operate. For instance, the bots engaged in the pro-vaccination network display the same behavior as that typically associated with bots disseminating health disinformation. This behavior is consistent with their own nature as message repeaters [14,18], increasing impressions upon a given message or facilitating interactions [26], clearly intending to influence human users’ opinion [15] on healthcare matters [10,11,12]. This behavior is mostly focused on disseminating information about vaccines by tweeting socially or politically tinged messages, either in the pro-vaccination or anti-vaccination network, a situation that is consistent with findings raised by authors, such as Broniatowski et al., 2018 [12].

Third, with respect to the dissemination of information commonly associated with bots in relation to health information, we must differentiate between the networks analyzed, although both networks agree that the main focus of the bots is the subject matter of political messages or messages not expressing opinions, including images or hashtags.

Regarding the anti-vaccination network, the messages mostly have an anti-vaccination approach against COVID-19, either through political or social content. They even fabricate news about false problems or side effects associated to vaccines [29]. In our study, the impact of the bots was of little relevance, both in terms of generating interactions and impressions in the network [12], in addition to continuing the dissemination of low-credibility health information [30]. In this network, human users are responsible for the dissemination of anti-vaccine information [16,31]. Although, in our study, the effect is limited, the consequences of bots spreading health disinformation in times of health emergency should not be disregarded [14].

Furthermore, we found that bots engaged in the pro-vaccination network elicit a significant impact on what is defined as interactions with the network. This finding is consistent with what was previously described about the characteristics of messages with a pro-vaccination approach; that is, they are networks in which user participation is promoted [14]. The ability of bots to generate these interactions could be associated with their behavior in the pro-vaccination network, not being mere repeaters of information [8] but generators of a higher level of engagement with the human user [16], by mimicking the behavior of humans [17]. It is important to consider that users distrust information coming from accounts clearly identified as bots [32]. Hence, social bots can generate changes in the individuals they interact with, as the information received is more trustworthy [30].

The influence of the bots in their respective networks was another element assessed in this study. It was observed that bots in the anti-vaccination network do not play an important role amongst the 20 most influential users in the network. This situation is consistent with what was observed in other studies, where the role of bots was not prominent in the activity of the networks analyzed [16,26,28,31]. However, in the pro-vaccination network, 2 of the 20 most influential users were categorized as bots, reaching scores of 0.84 in Botometer. These data, at least to the researchers’ knowledge, have not been observed in other studies analyzing the use of bots fostering public-health policies in social networks.

As a result of our research, we found a new line of work related to the use of bots by healthcare organizations and healthcare workers, who are one of the biggest disseminators of examples and information. They also encounter rejection and uncertainty about vaccines, due to a lack of adequate information, the speed, and validation of vaccines, etc. [33]. We must understand the importance of these tools and use them for the common benefit of society, as in public-health policies, but always with identifiable bots to be credible. The great demand for information brought about by the recent pandemic must be accompanied by a major effort in scientific dissemination, which is evidence-based and regulated to avoid ethical implications and much of this work can be done in an automated way by bots with advice from health workers and government organizations.

Our study has some limitations. First, the analysis is only focused on Twitter, thus, being limited to this social network. In addition, when retrieving information using specific hashtags and keywords, it is possible that we missed users who posted anti- or pro-vaccination messages related to COVID-19 without using these specific keywords and hashtags. Another one is related to the use of the Botometer, since this tool presents some difficulties in detecting hybrid-type bots [14], which could mean the number of bots in our study is underestimated.

## 5. Conclusions

To the authors’ knowledge, the present study is the first to analyze the presence of bots in two networks, one in favor and another opposing vaccination against COVID-19, considering Twitter posts written in Spanish.

We believe that the findings shown in this study display information of interest to health organizations, both to better understand the role of bots spreading health misinformation and to assess the potential use of tools to disseminate verified health messages and information on social networks.

Although correcting Twitter misinformation remains a huge public-health challenge, the utilization of bots designed and generated by health organizations could be an extremely effective tool to make health misinformation on social media fade away. However, also, given the data observed in this study, bots can be extremely useful for the dissemination of messages in favor of public-health policies, generating a greater scope of verified health information. Given the high degree of interaction amongst users, it may be beneficial to strengthen their views and confidence on public-health policies.

Finally, we believe that the constant and regular use of social bots by healthcare organizations, together with training policies for healthcare professionals in the use of social networks, can be of great interest to enhance their participation and improve the effectiveness of healthcare communication.

## Figures and Tables

**Figure 1 vaccines-10-01240-f001:**
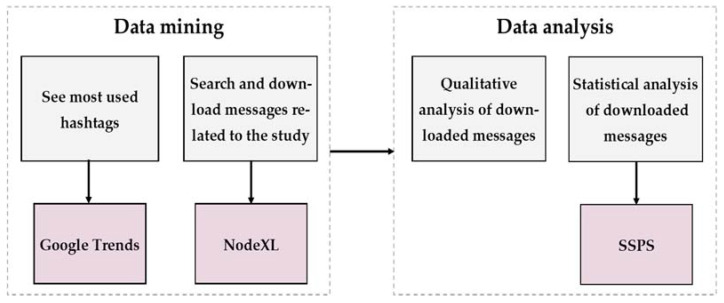
Data mining and analysis scheme.

**Figure 2 vaccines-10-01240-f002:**
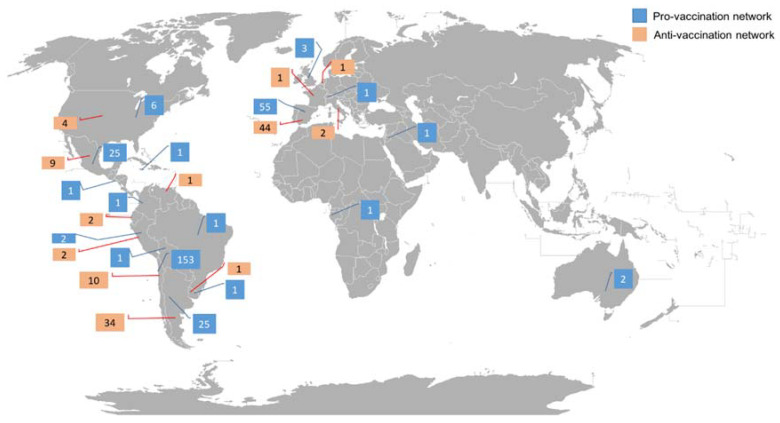
Bots’ location on pro and anti-vaccination networks. (Figure adapted from Fobos92′s world map with CC BY-SA 3.0 license).

**Table 1 vaccines-10-01240-t001:** Characteristics of users, impressions, and interactions in pro and anti-vaccination networks.

	Pro-Vaccination Network					
		N	%	Ratio (I/m)	Mean	Comparison Humans-Bots-t Student (*p*-Value)
Users	Human	2421	93.73			
	Bots	162	6.27			
	Total	2583	100			
Messages	Human	4840	94			
	Bots	309	6			
	Total	5149	100			
Users’ Interactions	Human	937,283	95.7	193.65	387.15	3.44 (0.04) **
	Bots	42,135	4.3	136.36	260.09	
	Total	979,418	100	190.21		
Users’ impressions	Human	96,811,449	95	20,0002.37	39,988.207	0.323 (0.746)
	Bots	4,569,946	5	15,718.27	28,209.543	
	Total	101,668,395	100	19,745.27		
	**Anti-vaccination network**					
		n	%	Ratio (I/m)	Mean	Comparison Humans-bots-t Student (*p*-value)
Users	Human	4243	91			
	Bots	420	9			
	Total	4663	100			
Messages	Human	70,263	90.61			
	Bots	7280	9.39			
	Total	77,543	100			
Users’ Interactions	Human	93,151	95.17	1.33	21.95	4.198 (0.001) **
	Bots	4724	4.83	0,65	11.24	
	Total	97,875	100	1.26		
User’s impressions	Human	5,571,594,549	92.37	79,296.28	1,313,126.22	0.22 (0.826)
	Bots	460,521,238	7.63	63,258.41	1,096,479.14	
	Total	6,032,115,787	100			

where: Ratio IM, means Ratio between number of impressions or interaction by message in the network and (**) denotes statistically significant differences.

**Table 2 vaccines-10-01240-t002:** Characteristics of most influential bots in pro and anti-vaccination networks.

Network	User Code	Description	Bot Score	BSC	Network Activity
Anti-vaccination	AV1	Citizen	0.78	16,444.45	Criticism to government
	AV2	Citizen	0.8	16,008.92	Conspiration: the vaccine as a means to foster genocide
	AV3	Citizen	0.78	15,228.38	Support to vaccine against COVID-19
	AV4	Citizen, nonconformist	0.78	14,001.76	Negationist: neither the virus nor the pandemic does exist
	AV5	Citizen	0.76	13,453.91	Criticism to government
Pro-vaccination	PV1	Political activist	0.84	91,7876.41	Spread of news about vaccines approvals
	PV2	Citizen	0.84	160,118.69	Information about vaccination set-off
	PV3	Political activist	0.88	76,177.66	Information about vaccination set-off
	PV4	Citizen	0.8	45,713.95	Approval of vaccines by the European Medication Agency
	PV5	Citizen	0.78	24,801.58	Spread of positive information on vaccines availability

where: BCS means Betweenness Centrality Score.

**Table 3 vaccines-10-01240-t003:** Categorization of messages in the pro- and anti-vaccination network movement.

Category	Pro-Vaccination Network	Anti-Vaccination Network
Political content	51.85% (84)	18.6% (78)
Vaccine awareness	11.11% (18)	
General tweets not expressing a view or opinion	29.63% (48)	33.02% (139)
Conspiracy theories		13.48% (57)
Pandemic negationism		3.72% (16)
Anti-vaccine tweets		26.97% (113)
Opposed to main subject of the network	7.41% (12)	4.21% (17)

## Data Availability

The data that support the findings of this study are available from the corresponding author upon reasonable request.

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
