# Peer review of "Bots’ Activity on COVID-19 Pro and Anti-Vaccination Networks: Analysis of Spanish-Written Messages on Twitter"

_vaccines, 2022, doi:10.3390/vaccines10081240_

Round 1

Reviewer 1 Report

The following are several changes that should be made to improve the manuscript.

As readers may not necessarily be familiar with social network research, the procedure for extracting data from social networks should be explained in a little more detail.

In Table 1, the p-values are presented inconsistently so that sometimes it is n.s., sometimes < 0.05, and other times a value that is not statistically significant is indicated. It is recommended in all cases to include the p, and if desired to use asterisks to show when there are statistically significant differences.

The ethical implications of the proposal that healthcare organizations use bots should be considered. Should they be identified as such?

Tables should be readable independently of the text. The abbreviations used should be included at the foot of the tables. In table 2, the meaning of BSC must be indicated.

On line 227, immediately before the discussion, there is a table with no title or numbering.

On line 136, the manuscript says, "The comparison between groups was performed with the parametric test. Please indicate the test, it is supposed to be the student's t-test, but it is not shown.

Author Response

Dear reviewer,

Thank you for review our article. Please find below our response (in red) to review comments and feedbacks. You can find the changes in the article marked in red to make it easier to identify.

1.- Reviewer: As readers may not necessarily be familiar with social network research, the procedure for extracting data from social networks should be explained in a little more detail.
Authors: Thank you for your comment, we have described the data extraction and analysis process with a figure to make it easier to understand.

2.- Reviewer: In Table 1, the p-values are presented inconsistently so that sometimes it is n.s., sometimes < 0.05, and other times a value that is not statistically significant is indicated. It is recommended in all cases to include the p, and if desired to use asterisks to show when there are statistically significant differences.
Authors: Thank you for your comment, we have modified the p-values to present them more homogeneously.

3.- Reviewer: The ethical implications of the proposal that healthcare organizations use bots should be considered. Should they be identified as such?
Authors: Thank you for your comment, it is very interesting, although it goes beyond the initial purpose of this study. However, it does open up a future line of work. We have added an answer to this question in the discussion section.

4.- Reviewer: Tables should be readable independently of the text. The abbreviations used should be included at the foot of the tables. In table 2, the meaning of BSC must be indicated.
Authors: Thank you for your comment, we have added this abbreviation at the foot of the table.

5.- Reviewer: On line 227, immediately before the discussion, there is a table with no title or numbering.
Authors: Thank you for your comment and highlighting this omission. We have added the identification and title.

6.- Reviewer: On line 136, the manuscript says, "The comparison between groups was performed with the parametric test. Please indicate the test, it is supposed to be the student's t-test, but it is not shown.
Authors: Thank you for your comment and for highlighting this omission. Yes, we used Students' t-test, as can be seen in table 1. We have included this correction in Data analysis subsection.

Reviewer 2 Report

Authors wrote on important topic in a COVID 19 pandemic

I appreciate it. 

Below my suggestions

1. Introduction: updata data on SARS CoV2 wordwilde at the day of resubmission

2. Methods and results: very well presented . Good both tables and figures

3. Discussion: add and discuss also the role of hesitancy in health worker as an important key to understand the hesitancy wordwilde (see and cite Attitudes towards Anti-SARS-CoV2 Vaccination among Healthcare Workers: Results from a National Survey in Italy. Viruses. 2021 Feb 26;13(3):371. doi: 10.3390/v13030371. )

4. Conclusion: give some global health proposal that came from your interesting data

Author Response

Dear reviewer,

Thank you for review our article. Please find below our response (in red) to review comments and feedbacks. You can find the changes in the article marked in red to make it easier to identify.

1.- Reviewer. Introduction: updata data on SARS CoV2 wordwilde at the day of resubmission
datos actualizados mundiales CoV2 del SARS en el día de la nueva presentación
Authors: Thank you for your comment, we have updated the global data, as of the date of the resubmission, on SARS CoV2.

2.- Reviewer. Methods and results: very well presented . Good both tables and figures
Authors: Thank you for your comment, it is very much appreciated by us.

3.- Reviewer. Discussion: add and discuss also the role of hesitancy in health worker as an important key to understand the hesitancy wordwilde (see and cite Attitudes towards Anti-SARS-CoV2 Vaccination among Healthcare Workers: Results from a National Survey in Italy. Viruses. 2021 Feb 26;13(3):371. doi: 10.3390/v13030371. )
Authors: Thank you for your comment, it is very interesting and opens up a future line of work. We have added an answer to this question in the discussion section and introduced the reference.

4.- Reviewer. Conclusion: give some global health proposal that came from your interesting data
Authors: Thank you for your comment. We have added a response to this question in the conclusions section as a new line of work.